# Real-Time Tracking of Human Neck Postures and Movements

**DOI:** 10.3390/healthcare9121755

**Published:** 2021-12-19

**Authors:** Korupalli V. Rajesh Kumar, Susan Elias

**Affiliations:** 1School of Electronics Engineering, Vellore Institute of Technology, Chennai 600127, India; 2Centre for Advanced Data Science, Vellore Institute of Technology, Chennai 600127, India; susan.elias@vit.ac.in

**Keywords:** inertial measurement unit, kinematic data, kinetic data, musculoskeletal disorders, neck movements, neck postures, OpenSim, random forest

## Abstract

Improper neck postures and movements are the major causes of human neck-related musculoskeletal disorders. To monitor, quantify, analyze, and detect the movements, remote and non-invasive based methods are being developed for prevention and rehabilitation. The purpose of this research is to provide a digital platform for analyzing the impact of human neck movements on the neck musculoskeletal system. The secondary objective is to design a rehabilitation monitoring system that brings accountability in the treatment prescribed, which is shown in the use-case model. To record neck movements effectively, a Smart Neckband integrated with the Inertial Measurement Unit (IMU) was designed. The initial task was to find a suitable position to locate the sensors embedded in the Smart Neckband. IMU-based real-world kinematic data were captured from eight research subjects and were used to extract kinetic data from the OpenSim simulation platform. A Random Forest algorithm was trained using the kinetic data to predict the neck movements. The results obtained correlated with the novel idea proposed in this paper of using the hyoid muscles to accurately detect neck postures and movements. The innovative approach of integrating kinematic data and kinetic data for analyzing neck postures and movements has been successfully demonstrated through the efficient application in a rehabilitation use case with about 95% accuracy. This research study presents a robust digital platform for the integration of kinematic and kinetic data that has enabled the design of a context-aware neckband for the support in the treatment of neck musculoskeletal disorders.

## 1. Introduction

Enabling technology to monitor, measure, and manage human movements has been an active area of research with a broad spectrum of applications ranging from medical diagnostics, rehabilitation, sports, fitness, behavior analysis, and gait-based bio-metrics. The historic landscape of research publications in the field has presented the use of vision-based (video cameras), sensor-based (Inertial Measurement Units IMUs), infra-red, and RADAR-based innovations to study human movement [1,2,3,4]. Quantitative analyses have traditionally been carried out using either kinetic or kinematic data for various applications. Kinematic data are obtained using IMUs and have been used in the study of musculoskeletal disorders (MSD) [5] and gait recognition. On the other hand, kinetic data provide details of the force of the component in motion and help to analyze the activation of muscles associated with the joint in motion [6]. Kinetic data are computed from the signals obtained from a Kinesiologic Electromyography (KEMG) device and are quantitative analyses for understanding muscle force and fatigue [7,8,9]. The motivation for the research presented in this paper was to design a robust methodology to integrate the kinetic and kinematic features for predictive analysis of human postures and movements. IMUs are Micro-Electro-Mechanical-systems (MEMs)-based devices that are widely used to develop wearable technologies [10]. IMU data have been integrated earlier with several proprietaries and open-source-based simulation platforms for analysis and visualization of movement data [11,12,13,14,15,16,17]. Here, the integration of IMU with OpenSim is presented, and it is currently a focus area in top research laboratories as well. OpenSim is a free, open-source simulation and modeling tool developed at Stanford University (https://opensim.stanford.edu/) (accessed on 9 November 2021). The built-in feature-extraction functionalities and contributions by the community make OpenSim a scalable and reliable tool for analyzing human movements. Besides presenting a step-wise procedure to integrate IMU data with OpenSim, this paper presents a novel methodology of combining kinetic and kinematic data for generating an insightful analysis of human movements. OpenSim provides details of muscle activation and supports joint modeling of kinetic and kinematic parameters. In the following subsections, the detailed methodology is presented. The results and discussions that follow will highlight the significance and applications of the proposed digital platform for movement analysis. 

### Motivation and Proposed Work

The goal of the proposed research is to present a methodology to measure and identify the postures of the human neck for the prevention and rehabilitation of musculoskeletal disorders of the cervical region. Improper neck postures due to sedentary lifestyles are a significant cause of cervical spine dysfunction in all age groups ranging from children to the elderly [18,19,20,21,22,23,24,25,26,27]. The neck region can also be affected due to improper neck postures that are inherent in certain kinds of occupations. Other lifestyle-related activities, including sleeping in the sitting position during travel, can trigger musculoskeletal problems around the neck. The motivation for this research was to propose a novel methodology to prevent disorders of the neck through timely detection and notifications [28]. A sensor-based Smart Neckband that can precisely detect neck postures was designed to monitor and generate alert messages as a preventive measure. This neckband can also be used to take measurements of the range of motion of the neck regions during therapy and rehabilitation [28]. Research works related to the use of sensors to track movements by obtaining kinematic data have been presented extensively in the literature under the field called Actigraphy [29,30]. These approaches have also been widely used for tracking movement through commercially available wrist-worn fitness monitors. However, to identify the postures of the neck, there are several challenges and limitations in only using kinematic data obtained using sensors. Hence, we explored the possibility of integrating kinetic and kinematic data for better accuracy in the detection of neck postures. In this paper, we present a robust integrated platform for the predictive analysis of human neck postures and movements using kinetic and kinematic data. In the following sections, the methods and materials used in this research are presented. 

## 2. Materials and Methods

There are various conventional methods for measuring or recording human neck movements. Neck Range of Motion (N ROM) measuring instruments can be designed with proximity sensors, and NROM can be calculated with local fixed points on the human face with respect to human nose as center point. From nose to ears, the distance can be calculated, and based on degrees of freedom, the NROM values can be obtained. Similarly, there is another method, using video and biomarkers; in this method, biomarkers on the human neck and simultaneous video recording can be used to track neck movements. In continuation of these methods, our proposed model will help to find human neck movements in the digital environment [31,32,33,34,35]. In this research, an IMU-based device is used for acquiring the kinematic data of the neck, and the OpenSim simulation modeling tool is used to generate the kinetic data of the corresponding neck movements. Predictive analysis to detect neck postures from the kinetic and kinematic data is performed using machine learning methods. Validation is shown using synthetic data. 

### 2.1. Kinematic Data Acquisition Using Smart Neckband

#### 2.1.1. IMU Neck Band

Inertial Measurement Unit (IMU) embedded in an elastic neckband captures the kinematic data required for the analysis. IMU devices are available in miniature sizes and can be used to design wearable products. For this research, Metawear CPRO, an IMU device developed at MBIENTLAB (https://mbientlab.com/metamotionc/) (accessed on 9 November 2021), has been used. It has on-chip memory, processing unit, accelerometer, gyroscope sensor, magnetometer sensor, pressure sensor, and temperature sensor. In addition to these sensors, this device has inbuilt Bluetooth support to establish communication with the associated mobile application or any Bluetooth device to transmit the device data. This IMU device can stream data for 8–24 h continuously using a 3.3 V coin-sized battery and is attached to a wearable band with an adjustable strap to fit it firmly around the neck. This neckband, when integrated with the proposed predictive analysis, can be referred to as a Smart Neckband for its context-aware functionalities. The primary use of this device is to record neck movements [28]. The technical specifications of the device and built-in sensors are given below:o Weight: 5.66 g;o Battery: 200 mAH coin battery;o Usage modes: 8–24 h (stream), 2–48 h (log);o Data Transfer: Bluetooth Low Energy Smart (BLE);o Flash Memory: 8 MB.o Built-in sensors:

Accelerometer: Range: ±2, ±4, ±8, ±16 g;Resolution: 16 bit;Sampling Rate: 0.001 Hz–100 Hz stream–800 Hz log.

Gyrometer:Range: ±125, ±250, ±500, ±1000, ±2000°/s, Resolution: 16 bit;Sampling Rate: 0.001 Hz–100 Hz stream–800 Hz log.

Magnetometer:Range: ±1300 μT (x,y-axis), ±2500 μT (z-axis);Resolution: 0.3 μT;Sampling Rate: 0.001 Hz–25 Hz.

#### 2.1.2. Mobile Application—MetaBase

In this research, an IMU-based device is used for acquiring the kinematic data of the neck, and the OpenSim simulation modeling tool is used to generate the kinetic data of the corresponding neck movements. Predictive analysis to detect neck postures from the kinetic and kinematic data is performed using machine learning methods. In the following sections, the methods and materials used in this research are presented.

#### 2.1.3. Sensor Data Format

The IMU device has built-in sensors to record various kinematic aspects of the movements. The format of the data captured by these sensors is given below:Accelerometer sensor: epoch(ms), -> time, elapsed(s), x(g), y(g), z(g);Gyroscope sensor: epoch(ms), -> time, elapsed(s), x(deg/s), y(deg/s), z(deg/s);Magnetometer: epoch (ms), -> time, elapsed(s), x(T), y(T), z(T), etc.

For the kinematic analysis of neck postures, dealt with in this research, the accelerometer sensor data were sufficient. The accelerometer sensor was operated at 100 Hz frequency with ±8 g. Figure 1 shows the kinematic data acquisition process using sensor system and MetaBase mobile application.

### 2.2. Kinetic Data Generation Using the OpenSim-Based Neck Musculoskeletal Model

Electromyography (EMG) and Surface EMG (SEMG) are the standard methods for muscle-related data acquisition for movement analysis. The kinetic data obtained from EMG-based methods are accurate but are limited to laboratory-based studies. To carry out movement analysis using both kinematic and kinetic data, a novel methodology has been proposed in this paper. Here, we focus on the detection of postures of the neck using an innovative approach. Instead of capturing kinetic data related to muscle activation using any of the EMG-based methods, in the proposed approach, kinetic data of the corresponding neck postures are generated using a Neck Musculoskeletal Simulation Model. The Smart NeckBand captures the real-world kinematic data of the neck postures, and these data are used to generate the corresponding kinetic data relating to the muscles around the neck region using the OpenSim simulation tool. 

#### 2.2.1. OpenSim—Simulation Modeling Tool and Its Features

This is the most popular open-source tool used to create and study human musculoskeletal models and provides extensive data on kinematics and kinetics of human movement. Built-in functionalities such as Scale, Inverse Kinematics (IK), Inverse Dynamics (ID), Residual Reduction (RR), Static Optimization (STO), Computed Muscle Control (CMC), and Analyze provide support to extract all information related to the muscles, tendons, joints kinetics, and kinematics. An overview of the functionalities of OpenSim and the inbuilt tools is shown in Figure 2, and detailed information is made available by the OpenSim contributors [36,37,38,39].

##### Built-In Tools and Its Features

**Scale:** In OpenSim, Scale is a built-in tool, which is used to create user-specific musculoskeletal models, mainly used to adjust the dimensions of the skeletal system in terms of Mass. In the Scale tool, we can adjust the Scale factors and static pose weights. In this tool, we must give marker data for measurement as input. From the scale, the output is the *.osim* file, which is the main source file.**Inverse Kinematics (IK):** In OpenSim, Inverse Kinematics is a built-in tool, which is used to generate the Inverse Kinematics of the musculoskeletal system concerning joint movements. The input for this tool is three-dimensional coordinate data which are in *.trc* file format (track row–column). This input file gives information about the joint movements with respect to time in three dimensions (x,y,z). Based on this input, the IK tool will generate the motion; we can observe this in GUI—Graphical User Interface in the OpenSim tool. The output of this tool is the *.mot* file, which consists of information about the joint’s motion concerning time.**Inverse Dynamics (ID):** The Inverse Dynamic tool (ID) is used to determine the net forces and torques of joints, which are responsible for movement generation. IK tool output, i.e., .*mot* file motion file and ground reaction forces data in the .xml format, are fed to the ID tool as input sources. ID tool will perform the mass-dependent acceleration functions and generate the forces based on the conventional F = ma equation. The output of the ID tool is Inverse Dynamics.sto (ID-State storage file).**Residual Reduction (RR):** It is a built-in tool that works like Forward Dynamics and uses a tracking controller to follow kinematics extracted from the IK tool—nothing but movements.**Static Optimization (STO):** This tool helps to obtain muscle forces and activations at each instance in time. For this tool, input will be a .mot file—motion file and generates muscle forces and activations the format of .sto.**Computed Muscle Control (CMC):** This tool is a major block in the OpenSim software, which computes muscle excitations, joint movements such as kinematics, and kinetics of each component present in the musculoskeletal model. This tool generates .sto files for muscles, joints, and ligaments—active and passive fibers, power, length, forces, accelerations, and positions, etc.**Analyze:** This tool helps to analyze the model based on its simulation. If the duration of the simulation is long in terms of time, the Opting Analyze tool is the best option compared to the CMC tool. This tool helps to obtain accurate results in less time on the already simulated use case.

#### 2.2.2. Neck Musculoskeletal Model

In this research, the neck musculoskeletal model developed by [40], shown in Figure 3, was used. This is a fully flexible model for head and neck movements and versatile compared to other models [41]. The neck region consists of cervical joints (C1–C7), sixty-four muscles, and various associated tendons and ligaments. The hyoid muscles play a vital role in supporting the neck movements and hence have a vital role in the proposed predictive analysis [40,42,43,44,45]. The neck musculoskeletal model used in this research integrated the hyoid muscles, and this was a big advantage for our experimental analysis. The kinetic data provided by OpenSim includes forces, length, power, and activation levels of joints, muscles, and tendons. With the neck musculoskeletal model and research insights provided by the team [40], we simulated kinetic data of the hyoid muscles for our research analysis. The hyoid muscles ***are Digastric, Geniohyoid, Mylohyoid, Stylohyoid, Sternohyoid, Thyrohyoid, Sterno_Thyroid,*** and ***Omohyoid,*** shown in green color in Figure 3.

### 2.3. Experiments and Research Analysis

In general, the human neck has three degrees of freedom: a horizontal plane, a vertical plane, and the rolling of the head. All other asymmetric movements of the neck are variations and combinations of these three fundamental movements. At any point in time, the neck will be in one of the following nine static positions called neck postures, or it can move in any random order between these nine postures [28]. The nine static positions or neck postures are mentioned below and shown in Figure 4:

The goal of the research presented in this paper is to design a methodology to detect neck postures by training the machine learning algorithms using kinematic and kinetic data. In this research, we used Random Forest, an ensemble learning method for prediction and classification. The Smart Neckband captures the real-world data and integrates it with the OpenSim simulation platform. To effectively capture the neck kinematics, an experimental study was first carried out to determine the ideal location for the IMU device. The IMU device was fixed onto the elastic band, and the neckband could be worn in a manner that located the IMU device either in front or at the back of the neck region. 

#### 2.3.1. Experimental Study for the Location of IMU

Initially, we approached *thirty* participants for this research work. As per the Declaration of Helsinki, we guided all participants and informed them about the research procedure. In this process, we raised a query related to participants’ health information, particularly about their neck/cervical problems. 

**Condition:** Participants should participate voluntarily and should not have undergone any surgery/treatment for the neck/cervical region.

With this statement, ***eighteen*** participants were withdrawn from their participation due to their neck/cervical treatment history. 

Among others, ***four*** participants were withdrawn during the research process due to personal reasons. 

Finally, **eight** volunteers participated actively. Before the beginning of this research, once again, we made sure that volunteers never underwent any surgery/treatment for their neck. All the volunteers gave written consent after being fully informed about the research procedure. All the information gathered was based on the ***Declaration of Helsinki***. Participants’ physical attributes were tabulated in Table 1.

The subjects wore the neckband and participated in the research study. The participants were asked to keep their necks in the nine static positions for a duration ranging from 1 min to 2 min based on their comfort level. IMU data were recorded for all participants for each of the nine neck postures with the sensor located at the FRONT side of the neck region and similarly for the BACK side of the neck. Figure 5a shows the kinematic and kinetic data extraction methods using the IMU device and OpenSim simulation model for both the front and back of neck locations. Details of the dataset are presented in Table 2. 

Datasets were pre-processed and modularized based on the time stamp provided by the sensor and labeled manually. The observations in the dataset were also validated using the video footage of the corresponding experimental study. The quantity of the dataset is good enough for the training and testing mechanism to predict the neck postures. From each subject, 1080-time frames of information were generated, i.e., 1080 rows of corresponding acceleration data were generated for nine positions. All together, eight subjects’ data were merged into a dataset, which consisted of 8640 × 5 [rows × columns] of data. As part of the pre-processing task, NaN’s (Not a Number) and NA’s (Not Available) were interpreted, outliers were removed, and data were normalized.

#### 2.3.2. IMU Data Integration with OpenSim

The IMU-based accelerometer sensor data format provides three-dimensional kinematic data (x,y,z).To export IMU kinematic data into the OpenSim simulation tool, mathematical and functional analysis is required. In OpenSim, a file with the extension .trc (track row–column) is used as an input file for the Inverse Kinematics (IK) tool, and this tool provides joint movement data as a motion file with the extension *mot’*.The neck-skeletal model has seven sets of markers around the skull and cervical region (four on the skull, one at the Sternum Jugular Notch, and two at the right and left acromioclavicular joints). The marker Sternum Jugular Notch (SJN) is located on the front side of the neck. The IMU-based kinematics data are mapped onto the x,y,z coordinates of SJN. The other markers are calibrated according to the functional movements. The .trc file contains the details of these markers, and it is the input file for the Inverse Kinematics (IK), and the motion file (.mot) is obtained as the output.The information in this .mot file is fed as input to the Computed Muscle Control (CMC) tool, which produces the data related to neck kinematics, kinetics, joints, muscles, forces, etc.The functional integration of IMU data and OpenSim is shown in Figure 5b. Available results were interpreted, outliers were removed, and data were normalized.

#### 2.3.3. Smart Neckband—Comfort Level

We opted for cotton-material-based neck supportive bands, which are commercially available in the market and flexible to fit around the neck with Velcro adjustments. Then, we integrated the IMU device with the neckband. After finishing the research, we collected feedback from the participants on the comfort level of wearing the Smart Neckband. Table 3 shows the feedback given by the subjects on wearing the Smart Neckband. Based on overall feedback, we concluded that wearing this Neckband did not create any kind of disturbance for the participants, and we strongly believe that they were happy to wear it; based on their satisfaction, they had given ratings.

### 2.4. Predictive Analysis Using Machine-Learning Algorithms

The data collected by keeping the IMU device at the FRONTSIDE of the neck were divided into training and testing data in the ratio of 75:25 and given as input to the machine-learning algorithms to classify the nine static neck positions and to find the accuracy of the classification. Similarly, the data collected by keeping the IMU device at the BACKSIDE of the neck were processed. 

#### 2.4.1. Machine-Learning Algorithms Used in This Research

**K-Nearest Neighbors (KNN):** KNN is a supervised learning algorithm, which calculates the nearest distance of a similar object; this is why it is sometimes called a proximity or closeness-finding algorithm [46,47]. 

**Decision Trees (Iterative Dichotomiser 3 (ID3)):** This is a supervised learning algorithm, which uses Information Gain values to decide important contributing features to classify the data [48,49]. 

**Random Forest Algorithm:** This is a supervised learning algorithm; it is a combination of multiple Decision Trees; this ensemble algorithm works for classification and regression problems [50,51]. 

#### 2.4.2. Algorithmic Responses—Result Analysis

Figure 6a shows the machine-learning algorithms’ responses. Among three algorithms, the Random Forest algorithm has shown significant results in terms of classification accuracy. Further, the Random-Forest-algorithm-based confusion matrix and performance metrics were generated for the front and back location and are shown in Figure 6b,c. A total of 100% accuracy was achieved for the front-location-based classification and 99% for the back-location-based classification. From this research study, *we can infer that the ideal location for the IMU device during data capture is the front side of the neck.* This inference correlates with our decision to use the hyoid muscles located on the front side of the neck for the accurate classification of neck postures. In this research paper, we proposed the idea of generating kinetic data related to the hyoid muscles and using this data along with the associated kinematic data to accurately detect neck posture using classification techniques. This is presented in the following section. 

## 3. Results

### 3.1. Robust Integration of Kinematic and Kinetic Data

In this section, we first present the procedure to integrate kinematic and kinetic data for the experiment analysis. Integrating the IMU-based kinematic data to the OpenSim simulation modeling platform is a challenging step in the research domain [28].

### 3.2. Subject-Specific Neck Postures

In this research, we recorded and analyzed kinematic data for all the subjects. In this section, we present the simulation-based neck posture of ***Subject ID: 04***. The data collected from all the other subjects were used in training and testing for the classification. Figure 7 shows the neck postures of ***Subject ID: 04*** and the corresponding musculoskeletal postures in OpenSim.

### 3.3. OpenSim—Neck-Musculoskeletal-Model-Based Kinematic and Kinetic Data Analysis

OpenSim generates neck kinematics information based on input data: acceleration, position, and velocity. We used the acceleration and position data to classify and predict the human neck posture. From Figure 7, we can observe the response of the cervical joints and other associated joints during the experimentation task.

The subjects changed the neck posture from one position to another after a time gap of about 120 sec, and with a total of nine positions in the study, 1080 s of data were captured. Figure 8a shows the variations in the neck acceleration concerning joint kinematics, and similarly, Figure 8b shows the variations in the position of the neck. OpenSim-based IK tool generates the movements (.*mot*) data, which are fed to the CMC tool as input and extracted the kinetic data as an output. From the output data, we analyzed the neck joint movements corresponding to neck positional changes. The CMC tool calculates the Body acceleration and position data.. From this data, we observed that few joints and muscles excite high for cetain neck movements, and few respond low for certain neck movements.From the corresponding figures, we can observe the changes in the neck joint’s momentum. 

There are eight important sets of hyoid muscles ***(***shown in Figure 3), and many other associated hyoid muscles are attached to the hyoid bone in the neck region. These muscles help in providing free movement generation and flexibility to the neck [40,52]. The OpenSim CMC tool provides kinetic information such as forces, activation, lengths, etc. Using the CMC tool, kinetic data were extracted for the corresponding kinematic data that were captured and integrated with OpenSim. Here, the response of the tendon forces of the neck hyoid muscles was analyzed, as shown in Figure 8c. 

### 3.4. Predictive Analysis—Kinematics and Kinetics

Earlier, we covered various research works carried out on posture classification methods. In this process, we did not find any potential research works similar to this research. In this respect, we observed that a few researchers performed posture prediction (hand gestures, body position, sitting, standing, squats, etc.) using Machine and Deep Learning methods. They considered sensors-, video-, and markers-based datasets for posture prediction. They achieved strong and accurate results using these methods [50,53,54,55,56,57,58,59]. In this research, we opted for Machine Learning algorithms for the prediction of neck postures/movements. 

We have observed the performance of the Machine Learning algorithms on the datasets (Section 2.4) based on the performance metrics of the algorithms. We opted for the Random Forest (RF) algorithm for posture prediction. RF was used to classify and predict the neck posture based on the kinematics and kinetics data generated by OpenSim. The Random Forest algorithmic approach achieved 100% accuracy in the classification of neck postures using neck acceleration and position data. These results are presented in Figure 9a,b. Similarly, the Random Forest classifier predicted nine neck postures using the response of the tendon force data of hyoid muscles and achieved 100% accuracy in the prediction. The result of the kinetic tendon force classification is shown in Figure 9c. 

## 4. Validation—Use Case Model: A Predictive Model for Rehabilitation Monitoring and Assessment Based on Neck Movements

### 4.1. Rehabilitation

Rehabilitation therapy for musculoskeletal disorders is normally prescribed by a certified physiotherapist. Based on the severity of the injury, the treatment can involve heat, cold, exercise, massage, and ultrasound methods. Exercise-based therapy is presented as a use case to demonstrate the novelty of the research proposed in this paper. 

### 4.2. Neck Movements

For the experimental study, the subject wore the neckband and performed the neck movements as an activity in a random sequence. The sequences of movements were randomly selected neck exercises performed over 2 min of the time frame. The subject randomly moved his neck and head from one position to the other, and the following sequence is one such instance: *NM-NL-NM -NRU -NM-NLD-NM-NU-NM-NR-ND -NM-NLU-NM* (abbreviations mentioned in Section 2.3). The sequences varied for every experimental trial. In this study, neck movements were captured using the Smart Neckband, and the video recording was also performed simultaneously for sensor data segmentation and validation purposes.

### 4.3. Rehabilitation Monitoring System—Methodology and Results

The entire workflow of the rehabilitation monitoring system is presented in Figure 10. The complete structure of the rehabilitation monitoring system consists of two stages.

***Stage 1:*** The workflow shows the details of how the proposed model is trained (based on Section 2.3.3).

***Stage 2:*** The trained model is used to identify the neck postures that define the movements.

The acquired sensor dataset consists of 1238 rows and 5 features (time, accelerometer (x,y,z), movement) and is collected for 2 min. The video that was simultaneously recorded was used to trim the dataset to align with the exact neck movement data. The neck movement data were collected and saved with corresponding neck movement labels. Forty-nine samples of kinematic data points of neck postures were segregated and labeled. These labeled neck movements were exported into the OpenSim simulation tool, and corresponding kinetic data were obtained. OpenSim-based built-in tools provide options for data-capturing relevant to joint kinematics, kinetics, active and passive fiber forces of muscles and tendons, activation, lengths, velocity, power, etc. In this research, active-force data of the hyoid muscle were obtained and used for further analysis. For training the model, an integrated kinetic and kinematic dataset was used. The dataset comprised of kinetic data of various neck static postures with labels, recorded by IMU, and corresponding muscle activation forces of hyoid muscles was obtained from OpenSim.

### 4.4. Observations

The model was trained using the Random Forest algorithm [18], which was used to predict the classes of neck postures. Here, the output of neck movements is predicted in a sequence based upon time frames. Figure 11 shows that there are nine classes mapped in a circular shape. Each class represents one static neck posture point. The neck movement from one posture to another is indicated with directed arrows and is labeled in Figure 10. 

Actual Movements: NM-NLU-NM-ND-NR-NM-NU-NM-NLD-NRU-NM-NL-NM.

**Instance 1:** Actual movements show that the neck movement started from the Neck Middle position to Neck Left Up and then came back to Neck Middle. Then, it moved from Neck Middle to Neck Down, then to Neck Right, and from there to Neck Middle. Then, it moved from Neck Up and then back to Neck Middle and then towards Neck Left Down. From there, it moved to Neck Right Up then came back to Neck Middle. Finally, it moved to Neck Left and came back to the Neck Middle position. The kinetic values of the force of hyoid muscles during the movements were used as the training data. 

**Instance 1:** NM-NLU-NM-ND-NR-NM-NU-NM-NLD-NRU-NM-NL-NM.

**Predicted 1:** NM-NLU-NM-ND-NR-NM-NU-NM-NM-NRU-NM-NL-NM.

Predicted movements show that ***93.33%*** accuracy was achieved, and one neck movement was wrongly predicted.

Similarly, as a test case, we verified different neck movements for different instances. For instance, **10**, we obtained ***100%***accuracy. 

**Instance 10**: NM-NR-NL-NM-NR-NM-ND-NM-NU-NM-NR-NU-NM-NLD-NM.

**Predicted 10**: NM-NR-NL-NM-NR-NM-ND-NM-NU-NM-NR-NU-NM-NLD-NM.

### 4.5. Experimental Use Case—Rehabilitation Scoring System Using Synthetic Data

As a use case, a synthetic dataset was created to validate the proposed method. In this process, the dataset was created based on sample neck movements related to basic exercise patterns. These data were used to monitor and validate the rehabilitation scoring system. Figure 12 shows a 10-day neck movement monitoring and validation process.

In this process, actual neck movements were recorded as **NM-NM-NU-NU-NU-NU-NU-NU-NM-NM-ND-ND-ND-NM-NM-NU-NU** in a timeline. As indicated in Figure 11, these movements were compared with day-wise trails. In a day, two trials were conducted, and data were compared with actual movements. Based on these movements, rehabilitation scoring can be calculated. In Table 4, the rehabilitation monitoring and assessment report is summarized. Day-wise improvements are mentioned in Table 4. This result shows the effectiveness of the proposed methodology. The scale for the assessment depends on the rehabilitation scoring system shown in Figure 12. 

### 4.6. State of the Art: Objective vs. Research Flow

The presented research work should fill the gap between conventional physio assistive devices and technology. Recent advancements in artificial intelligence methods and hardware devices can bring a novelty in the physiotherapy processes. In this aspect, this research work can be the first approach to bring automation to the physiotherapy and assessment system. 

**Obj. 1:** The main purpose of this research is to provide a digital platform for analyzing the impact of human neck movements on the neck musculoskeletal system. 

**Obj. 2:** The second objective was to enable remote access to the therapist and to design a rehabilitation monitoring system that brings accountability to the treatment prescribed.

Research Flow: Figure 13 shows the entire research flow, which signifies research object 1, which highlights building a digital platform for analyzing the human neck postures and movements and their impact on the musculoskeletal system.

This research theme is offline-process-based, where we have to obtain IMU-based neck movement data and manually have to feed them into OpenSim software to extract the kinematics and kinetic data. These data are supplied to an AI engine; it predicts the postures/movements. Based on movements observations, the physiotherapist can analyze the patient’s condition. In terms of automation, these movement data feed to the AI engine, which can predict the posture/ movements changes and generate the assessment status. Based on this report, physiotherapists can analyze the patient’s condition and give appropriate treatment or therapy. Based on technology limitations, we have performed the entire research using an offline process. Based on advancements, we can fulfill Obj. 2 in the future. 

### 4.7. Future Scope

Advancements in technologies such as Machine Learning, Deep Learning, and Wearable Technologies will help to bring this innovation into the limelight in physiological measuring instrumentation. As for this research, the Smart Neckband for real-time tracking of human neck movements can be helpful to assist physiotherapists in rehabilitation. Based on limitations, we have performed this research using an offline process; in the future, advancements in technology can help to build a module that works in an online mode. Based on the online working process, we can monitor patient conditions remotely. We have performed significant work on a remote monitoring system for posture/movement prediction [28]; this work can be extended to building a remote rehabilitation system that will help physiotherapists in monitoring patients in a good manner. 

## 5. Conclusions

This paper presents a novel methodology to identify neck postures using kinetic and kinematic data. Improper neck postures can lead to cervical neck pain and musculoskeletal disorders. This research includes the design of a Smart Neckband consisting of an IMU device that captures kinematic data of the neck postures and movements. The OpenSim simulation tool and a neck musculoskeletal model were used to simulate the related kinetic data for the classification of neck postures. The Machine Learning algorithms achieved 100% accuracy in the prediction of neck postures. In addition to this concept, an evidence-based novel methodology is proposed for the prediction of neck movements to monitor the therapy of neuro-musculoskeletal neck disorders or injuries. Kinematic and kinetic data were integrated innovatively and used to train a model using the Random Forest algorithm. A motivating use case was presented, and this application helped to increase the potential of this innovation. The novel methodology proposed in this paper allows patients to observe their neck movements and exercise patterns to understand how specific exercises help in recovery from musculoskeletal injury. The proposed-technology-enabled system provides valuable insights to physiotherapists in understanding the progress of the patient’s condition. The future scope of this research is to embed the entire research work in a single device; this can enable the therapist to have remote access and analyze the human neck movements in an online mode. It also brings in the much-needed accountability to verify if patients are following the recommended therapy. This rehabilitation monitoring mechanism can also be used for remote assessment of musculoskeletal disorders.

## Figures and Tables

**Figure 1 healthcare-09-01755-f001:**
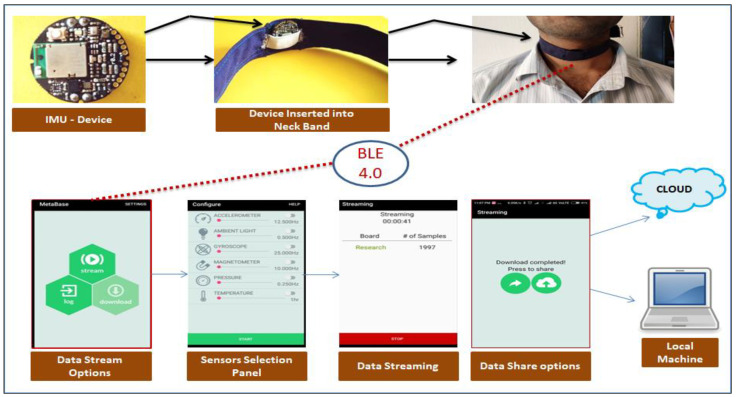
Kinematic data acquisition process.

**Figure 2 healthcare-09-01755-f002:**
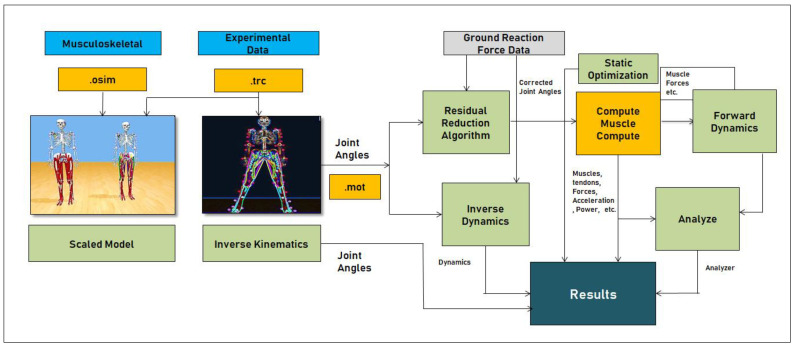
Overview of OpenSim functionalities.

**Figure 3 healthcare-09-01755-f003:**
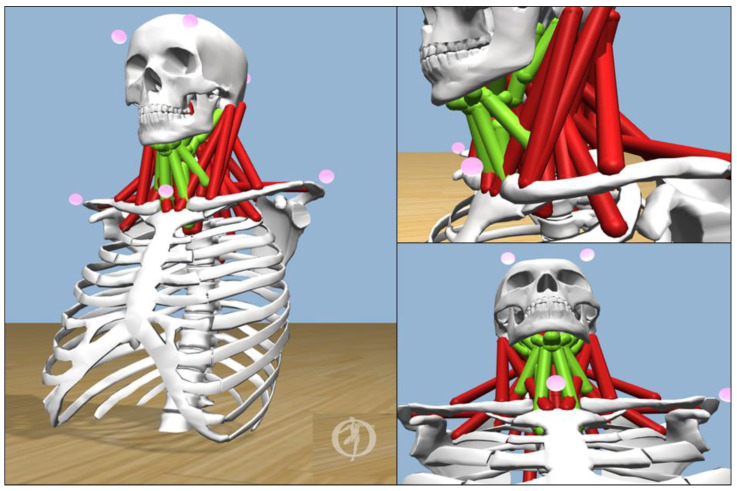
Hyoid muscles (in green color).

**Figure 4 healthcare-09-01755-f004:**
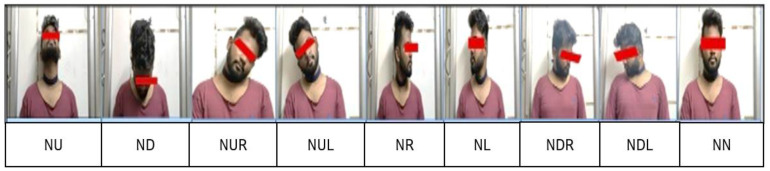
Neck Postures (Nine Positions)—Subject. 1. Neck at Extreme Up (**NU**), 2. Neck at Extreme Down (**ND**), 3. Neck at Extreme Right (**NR**), 4. Neck at Extreme Left (**NL**), 5. Neck at Right Up (**NRU**), 6. Neck at Right Down (**NRD**), 7. Neck at Left Up (**NLU**), 8. Neck at Left Down (**NLD**), and 9. Neck in the Middle (**NM**).

**Figure 5 healthcare-09-01755-f005:**
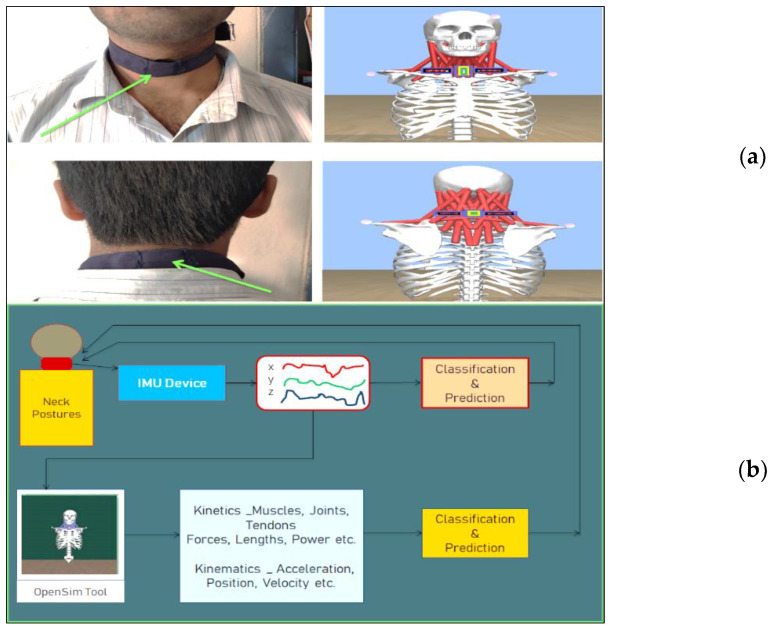
(**a**) IMU locations and corresponding musculoskeletal model; (**b**) IMU to OpenSim—data flow.

**Figure 6 healthcare-09-01755-f006:**
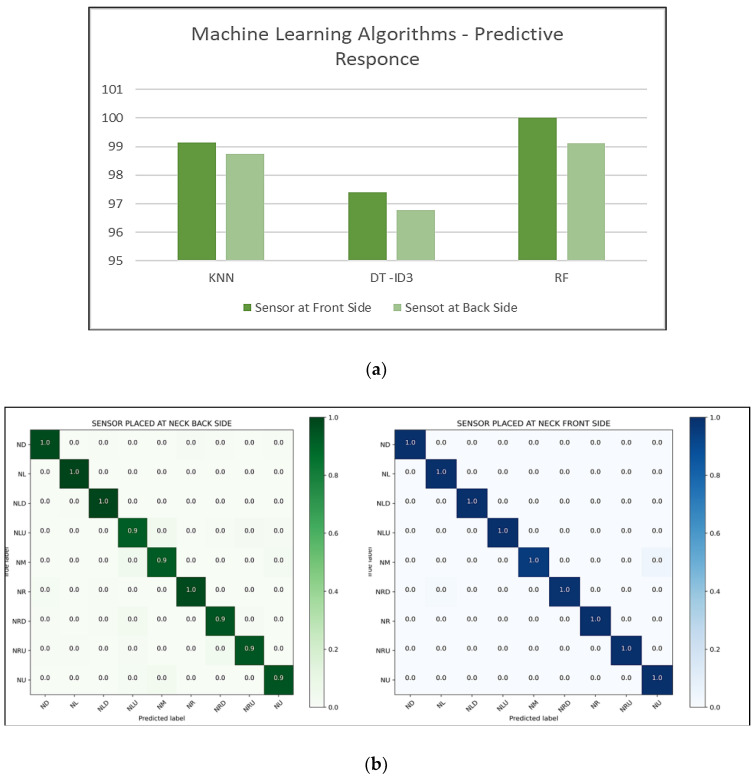
(**a**) Machine Learning algorithms—accuracy of the classification models; (**b**) confusion matrix—classification of neck posture based on sensor position; (**c**) performance metrics.

**Figure 7 healthcare-09-01755-f007:**
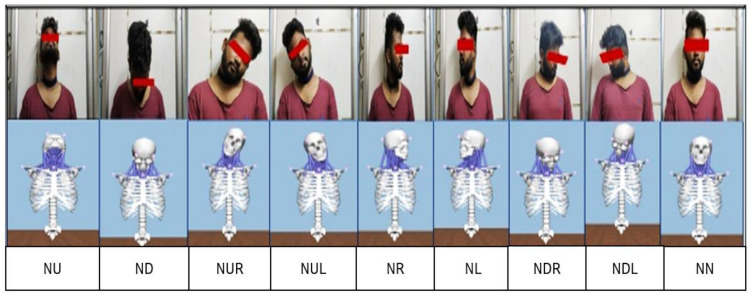
Subject-specific neck postures with corresponding musculoskeletal postures in OpenSim.

**Figure 8 healthcare-09-01755-f008:**
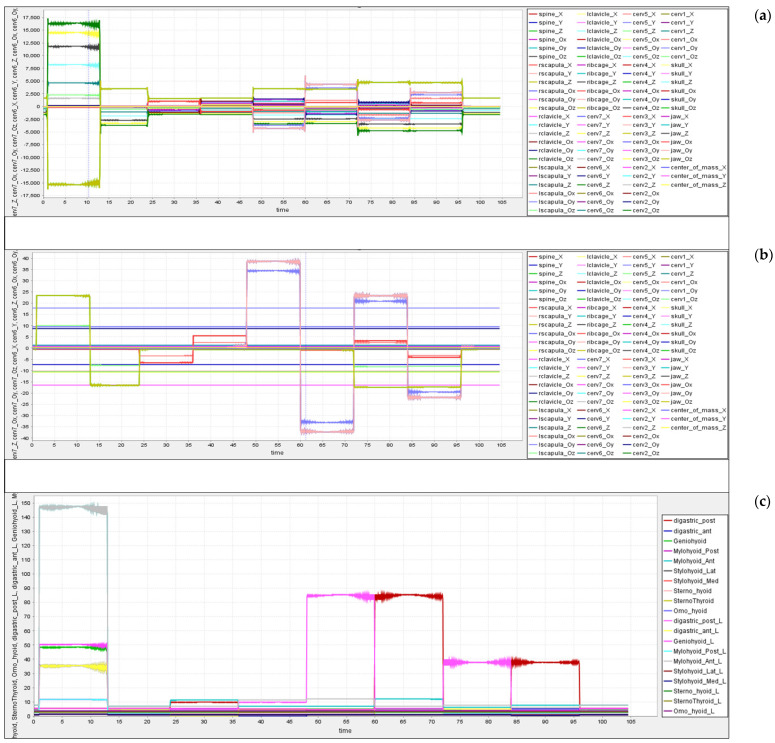
(**a**) Kinematics—neck acceleration; (**b**) kinematics—neck position; (**c**) kinetics—tendon forces.

**Figure 9 healthcare-09-01755-f009:**
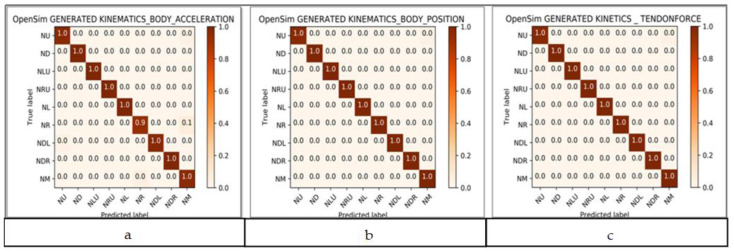
(**a**) Kinematics—neck acceleration; (**b**) kinematics—neck position; (**c**) kinetics—tendon forces.

**Figure 10 healthcare-09-01755-f010:**
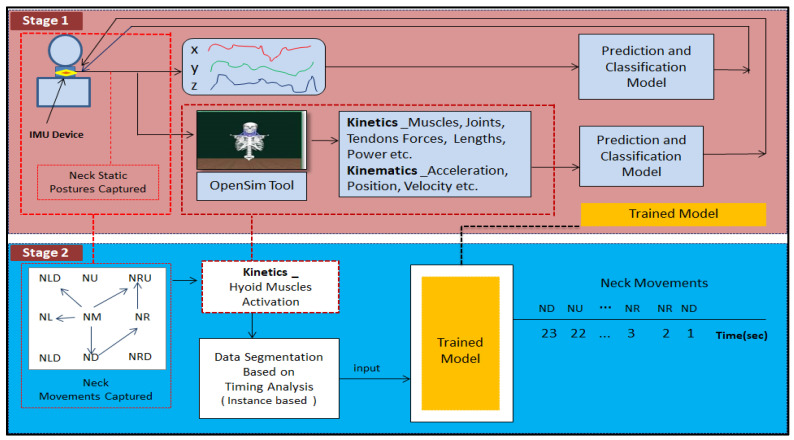
Block diagram of the research workflow.

**Figure 11 healthcare-09-01755-f011:**
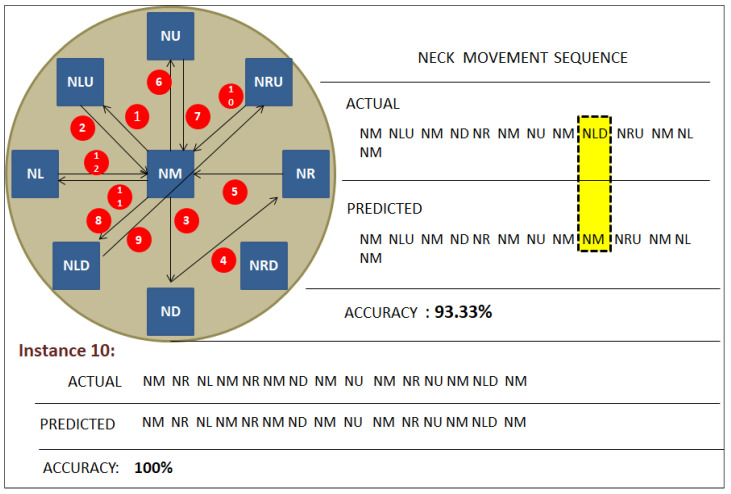
Rehabilitation scoring system.

**Figure 12 healthcare-09-01755-f012:**
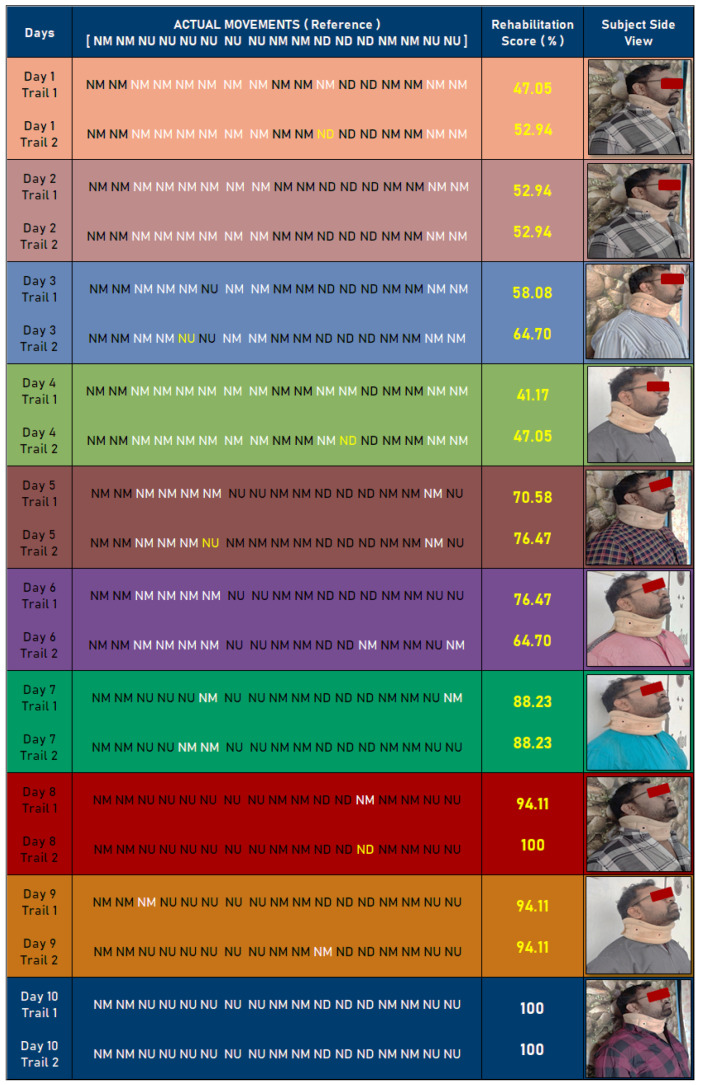
Synthetic dataset-based analysis of rehabilitation system.

**Figure 13 healthcare-09-01755-f013:**
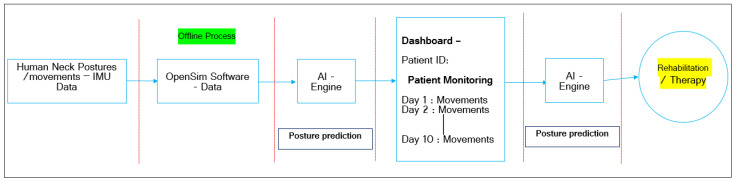
Research flow.

**Table 1 healthcare-09-01755-t001:** Volunteers’ physical attributes.

Volunteers = 8		
Attributes	Means +/− SD	Range
Age, year	39.5 +/− 17.6	27–52
Mass, kg	71 +/− 9.8	64–78
Height, m	1.45 +/− 0.09	1.38–1.52
Gender M/F	5/3	

**Table 2 healthcare-09-01755-t002:** Dataset quantitative parameters.

IMU—Device Placement	Total Time Duration	Dataset (After Pre-Processing)
A sensor placed at the front side	1080 s for nine static positions	1080 × 5 time, acc(x,y,z), position
A sensor placed at the back side	1080 s for nine static positions	1080 × 5 time, acc (x,y,z), position

**Table 3 healthcare-09-01755-t003:** Participants’ feedback on wearing Smart Neckband.

Condition	Sub 01	Sub 02	Sub 03	Sub 04	Sub 05	Sub 06	Sub 07	Sub 08
Any itching around the neck?	X	X	X	X	X	X	X	X
Is it comfortable to wear?	**✓**	**✓**	**✓**	**✓**	**✓**	**✓**	**✓**	**✓**
Is it easy to adjust to your neck dimensions?	**✓**	**✓**	**✓**	**✓**	**✓**	**✓**	**✓**	**✓**
Is there any trouble/pain/discomfort from wearing this Neckband?	X	X	X	X	X	X	X	X
Please give a rating for this Smart Neckband out of 5	4	4	3.5	4.5	5	3.5	5	4.5

**Table 4 healthcare-09-01755-t004:** Rehabilitation monitoring and assessment—model analysis report.

No. of Days	No. of Trails	Rehabilitation Assessment Based on Movements
1	1	Problem: Neck Up movement needs to improve
	2	Problem: Neck Up movement needs to improve
2	1	Problem: Neck Up movement needs to improve
	2	Problem: Neck Up movement needs to improve
3	1	Problem: Neck Up movement— slightly improved compared to previous day
	2	Problem: Neck Up movement—same as the previous trial
4	1	Problem: Neck Up and Neck Down movements— needs to improve
	2	Problem: Neck Up movement needs to improve
5	1	Problem: Neck Up movement. Slightly improved
	2	Problem: Neck Up movement. Good Improvement
6	1	Good Improvement
	2	Good Improvement
7	1	Good Improvement
	2	Good Improvement
8	1	Good Improvement
	2	Good Improvement
9	1	Improved
	2	Improved
10	1	Improved
	2	Improved

## Data Availability

Not applicable.

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
