# Peer review of "Real-Time Tracking of Human Neck Postures and Movements"

_healthcare, 2021, doi:10.3390/healthcare9121755_

Round 1

Reviewer 1 Report

The current form is become better than the original one. I have no more question. 

Reviewer 2 Report

The purpose of the device shown on this paper still remains unclarified. Several aspects are explained repeatedely ("the purpose is to provide a digital platform for analyzing the impact of human neck movements on the neck musculoskeletal system", "enable remote access to the therapist", "design a rehabilitation monitoring system"), but it is not explained how the device is achieving these objectives by detecting postures of the neck.

There is no state of the art that explains the current work in comparison with previous relevant scientific publications. Even when there is no similar work published, it would be important to highlight the innovative or differential aspects of the work presented compared with previous publications or research lines from others, regarding neck postures or movements detection.

In the Rehabilitation Assessment based on Movements (table 3) there are indications or feedback as Need to Improve, Slightly Improved, Improved or Good Improvement, and now there is information about the criteria used to scale that feedback or report. But there is no information about who decided or defined the scale, so there is no information about if this scale is clinically significant and would reflect a real recovery by the patient. That would be an important point if the device is aimed to be used as support for rehabilitation.

Conclusions are poorly explained, and several sentences are repeated from previous sections of the paper. There is a lack of information about the potencial of the device presented and analyzed, and the relationship with the aims previously explained.

Reviewer 3 Report

  • Figure. 9  is all 100%. This raises the question of whether the test dataset was small or we do NOT need a machine-learning algorithm to detect it; we can use a simple code to program it.
  • Otherwise, the paper is exciting. 

Round 2

Reviewer 2 Report

The paper presented has improved and the questions that arose were adequately answered. The aim of the research and its potential is clearer.

This manuscript is a resubmission of an earlier submission. The following is a list of the peer review reports and author responses from that submission.

Round 1

Reviewer 1 Report

This is a very interesting research and relevant to the healthcare journal. However, the improper organization and unclear statements increased the difficulty for the audience.

  1. some obvious mistakes need to correct, and these really showed that the quality of the research is questionable. For example, the sentence “in this research, an IMU-based device is used for acquiring the kinematic data of the neck……In the following sections, the methods and materials used in this research are presented.” is repeated. Moreover, the subtitle of Section 3.1 and 3.2 are the same.
  2. Section 2.2.1. The information about the OpenSim has lacked. 
  3. Section 2.3. The nine neck postures were selected in the study. How to identify the nine postures? Why were the nine neck postures used? What are the definitions? 
  4. The data collected from 8 participants were used for machine learning. Is it enough?
  5. The reason for applying Random Forest Algorithm for prediction is needed. Why not apply other methods?
  6. The detailed information related to data analysis should be addressed clearly.
  7. For the Discussion section, this is more like a validation. I can not find any discussion based on the results of the study. 
  8. The current study showed that the accuracy of the proposed method is 95%. These numbers cannot really present whether the proposed method is better than other methods. Comparison with other methods is required. These issues should be addressed to make it technically sound. 

Reviewer 2 Report

This study aims to develop a sensor-based real-time monitoring system for neck movements. The idea is original and the design could provide some future applications for rehabilitation.

It is certainly a current issue, and the work is interesting and innovative, as it concerns the use of a system developed to real-time monitor rehabilitation exercises for neck disorders  or to use it for prevention.

Nevertheless, I have some concerns pointed out below:

The ultimate aim of the device is not clear nor consistent. It is explained at the abstract that “the purpose is to provide a digital platform for analyzing the impact of human neck movements on the neck musculoskeletal system”, but then that aspect is no longer mentioned. It is also stated at the abstract that the objective is to enable remote access to the therapist to a rehabilitation monitoring system.

But then, at II. Motivation and proposed work, the purpose seems to be to prevent disorders through timely detection and notifications, and tracking movements during rehabilitations appears as a secondary outcome.

There is no state of the art that explains the current work in comparison with previous relevant scientific publications. It would be important to highlight the innovative or differential aspects of the new work carried out compared to the previous work.

It is stated that improper neck postures due to sedentary lifestyles are a significant cause of cervical spine dysfunctions in all age groups. This is a sentence without bibliographic reference, and in fact this issue currently in question, as there is not enough scientific evidence to support that posture is the cause of musculoskeletal disorders.

The sensor is attached to a wearable band firmly adjusted to the neck. That makes me wonder if it would be comfortable to the patients that are supposed to use it during their rehabilitation exercises, and there is no mention about this in the paper. Additional feedback from clinicians and/or patients could be useful in this sense, to assess the effectiveness and acceptability of the device, and therefore of its real future utility.

Physiotherapist work, health professional mention in the article, is poorly described and without information of their role on therapeutic exercise, that would be the main utility of the device presented.

Authors use the Hyoid muscles as a reference, as they play a vital role in supporting the neck movements. But there is no bibliographic reference that supports that they are vital in the predictive analysis. Is there previous work that supports this?

In the Rehabilitation Assessment based on Movements (table 3) there are indications or feedback as Need to Improve, Slightly Improved, Improved or Good Improvement, but there is no information about the criteria used to scale that feedback or report. There is neither information about who decided or defined the scale, so there is no information about if this scale is clinically significant and would reflect a real recovery by the patient.

It is too early to conclude that it is useful for the rehabilitation. Additional design works and experiments are required to improve the system and its readiness, and that should be mentioned in the paper.